# Supporting Relational Database Joins for Generating Literals in R2RML

Christophe Debruyne[1]

[1]*University of Liege – Montefiore Institute, 4000 Liège, Belgium*

### Abstract

Since its publication, R2RML has provided us with a powerful tool for generating RDF from relational data, not necessarily manifested as relational databases. R2RML has its limitations, which are being recognized by W3C's Knowledge Graph Construction Community Group. That same group is currently developing a specification that supersedes R2RML in terms of its functionalities and the types of resources it can transform into RDF–primarily hierarchical documents. The community has a good understanding of problems of relational data and documents, even if they might need to be approached differently because of their different formalisms. In this paper, we present a challenge that has not been addressed yet for relational databases–generating literals based on (outer-)joins. We propose a simple extension of the R2RML vocabulary and extend the reference algorithm to support the generation of literals based on (outer-)joins. Furthermore, we implemented a proof-of-concept and demonstrated it using a dataset built for benchmarking joins. While it is not (yet) an extension of RML, this contribution informs us how to include such support and how it allows us to create self-contained mappings rather than relying on less elegant solutions.

### Keywords

R2RML, Knowledge Graph Generation, Outer-joins, Joins

## 1. Introduction

R2RML [1] is a powerful technique for transforming relational data into RDF and was published almost a decade ago. R2RML was conceived for relational databases, but can be applied to relational data. Since then, it inspired many initiatives to generalize this approach for other types of data such as RML [2] and xR2RML [3]. Others looked at extending aspects of (R2)RML not pertaining to the sources being transformed, but to tackle unaddressed challenges and requirements such as RDF Collections [3, 4] and functions [5, 6].

The R2RML Recommendation specified a reference algorithm in which relational joins (natural joins or equi-joins, to be specific) can be used to relate resources. The implementation can be broken into two parts: (1) the generation of triples based on a triples map $tm_1$ related to a logical source, and (2) the generation of triples relating subjects from $tm_1$ with those of another triples map $tm_2$. While (2) does not use an outer-join, the combination of both (1) and (2) ensures that the data being transformed "behaves" as the result of an outer-join. The problem, however, is

*Third International Workshop On Knowledge Graph Construction Co-located with the ESWC 2022, 30th May 2022, Crete, Greece*

✉ c.debruyne@uliege.be (C. Debruyne)

🆔 0000-0003-4734-3847 (C. Debruyne)

CEUR Workshop Proceedings (CEUR-WS.org)

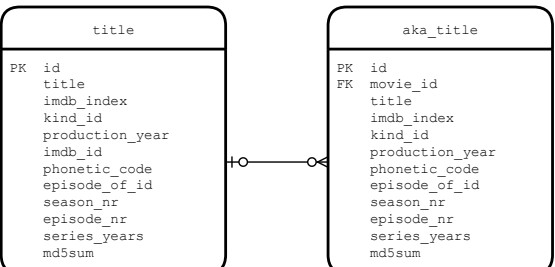

**Figure 1:** The tables `title` and `aka_title` of the database. A title may be related to one or more `aka_titles`, and an `aka_title` may be related to one `title`.

that support for such outer-joins is only limited to resources; there is no *convenient* way to do something similar for literals.

This paper proposes a simple extension of R2RML for supporting joins for the generation of literals. It furthermore proposes how the reference algorithm should be extended. We demonstrate this extension using a fairly big relational databases developed for bench marking joins [7]. This benchmark provides us also with a realistic case, motivating the need for such an extension. This paper furthermore positions this contribution with other initiatives developed by the Knowledge Graph Generation community with the aim of opening a discussion.

## 2. The Problem

We framed the problem in the previous section. In this section, we will rephrase the problem and discuss several approaches to achieve the desired result that one can observe in practice. To this end, we will be using a running example based on the database developed by [7].

To benchmark the performance of joins, [7] developed a database based on the Internet Movie Database[1] (IMDb). In short, their motivation was that existing synthetic benchmarks may be biased and real and "messy" provided better grounds for comparison. While that aspect is not important for this paper, the database they developed did contain two big tables: `title` containing information about movies and their titles, and `aka_title` containing variations in titles (either an alternative title, or titles in different languages). Figure 1 depicts the relation between the two tables and their attributes.[2]

There are two approaches to solving this problem with R2RML:

**Sol1** The first is the creation of two triples maps with one dedicated to the generation of triples for the outer-join. The problem with this approach is that the mapping is not self-contained and that there are two distinct triples maps which need to be maintained. One

---

[1]https://www.imdb.com/

[2]The files were loaded into a MySQL database, but required some minor pre-processing: a handful of encoding issues in the files and NULL values in aka_table were represented with the number 0. We also introduced a foreign key constraint that was not present in the SQL schema provided by [7]. The reason being that the foreign key constraint optimizes joins on these two tables. The tables contain 2528312 and 361472 respectively. There are 93 records from aka_table not referring to a record in `title` and 2322682 records in `title` have no alternative titles.

also needs to document that this construct was necessary to facilitate this outer-join. The advantage is that there are two distinct processes for querying the underlying database and thus less overhead.

**Sol2** The second, more naïve approach, is the use of one triples map with a (outer-)-join in its logical table. While this makes the triples map self-contained, unlike the approach above, but may require the processor to process many logical rows that generate the same triples.

We may observe, in the wild, cases of the first also being conducted for referencing object maps, especially when the processor used uses the reference algorithm. The problems with respect to self-containedness of triples maps still holds. An R2RML processor may internally "rewrite" referencing object maps as triples maps to optimize the process.

In the next section, we propose a small extension of R2RML to provide support for joins on literal values.

## 3. Proposed solution

In Listing 1, we demonstrate the extension. It introduces the predicate `rrf:parentLogicalTable`.[3] The domain of that predicate is `rr:RefObjectMap` and the range is `rr:LogicalTable`. Our extension requires that a `rr:RefObjectMap` must have either a `rrf:parentLogicalTable` or `rr:parentTriplesMap`. A referencing object map may now also generate literals. Where necessary, we will refer to object maps with a parent-triples map as "regular" referencing object maps.

```
1   <#title>
2       rr:logicalTable [ rr:tableName "title" ] ;
3       rr:subjectMap [ rr:template "http://data.example.com/movie/{id}" ; rr:class ex:Movie; ] ;
4       rr:predicateObjectMap [ rr:predicate ex:title ; rr:objectMap [ rr:column "title" ] ; ] ;
5       rr:predicateObjectMap [
6           rr:predicate ex:title ;
7           rr:objectMap [
8             rr:column "title" ;
9             rrf:parentLogicalTable [ rr:tableName "aka_title" ] ;
10            rr:joinCondition [ rr:child "id" ; rr:parent "movie_id" ] ;
11          ] ;
12      ] ;
13    .
```

Listing 1: Using parent-logical tables for managing joins

The reference algorithm[4] is extended as follows: step 6 will now iterate over all referencing object maps with a `rr:parentTriplesMap` and we add a 7th step for each referencing object map that uses a parent-logical table. The steps for generating are mostly the same. The two differences are: 1) it may generate any term type, and 2) the column names referred to by the

---

[3]The namespace `rrf` refers to the namespace used in [6].
[4]https://www.w3.org/TR/r2rml/#generated-rdf

object map are those of the parent. In other words, if both logical tables share a column X, then a reference to X would be to that of the parent. This behavior is consistent with that of regular referencing object maps. An implementation of this algorithm is made available.[5]

## 4. Demonstration

We now present a limited experiment comparing the performance of Sol1, Sol2, and our proposal using the relational database introduced in Section 2. The mappings for Sol1 and Sol2 are in Appendix A. In this experiment, we join using the tables as a whole. As R2RML requires result sets to have unique names for each column, we created a third table `aka_title2` where each column received the suffix '2'. We also created a foreign key from `aka_title2` to `title`. We wanted to avoid using subqueries to rename the columns, and these may become materialized and thus have an unfairly negative impact on the outcome.

The experiment was run on a MacBook Pro with a 2.3 GHz Dual-Core Intel Core i5 processor and 16 GB 2133 MHz LPDDR3 RAM. The database was stored in a MySQL 8.0 database in a Docker container. The code for the experiment was written in Java and ran the result of each mapping 11 times, of which the first run was removed to avoid bias from a cold start. The code calls upon the extension of R2RML-F and registered timestamps before and after executing the mapping. We have not registered the time for writing the graph onto the hard disk.

From Figure 2, which shows the average run times in seconds, it is clear that the approach of using two different triples maps (Sol1) is much faster than the two other approaches, which comes as no surprise. The problem, however, is that we have two distinct triples maps and their relationship is not explicit. Placing the outer-join in the logical table (Sol2) has the worst performance. The outer join yields a result set with 155749 more records than the referred table and contains twice the number of attributes. The overhead can be significantly reduced by only selecting the columns of interest, but the three mappings refer to the logical tables as a whole. Unsurprisingly, our solution is less efficient than Sol1 but considerably more efficient than Sol2.

We may conclude from these initial results that the proposed solution is not only a viable solution. It also ensures that the mappings remain self-contained. While performance is crucial in knowledge graph generation, we argue that even the vocabulary is a contribution and that an R2RML processor can rewrite referencing object maps (both types) into distinct triples maps.

## 5. Discussion

In this paper, we extended the concept of `rr:RefObjectMap` to support joins for literal values. The reference algorithm for R2RML processes these in a separate loop for the generation of relations between subjects of two triples maps. Our approach added a similar step to the generation of literals based on a join. One may ask whether this approach may be adopted for term maps in general. The generation of subjects, predicates, and graphs for relational databases is based on a logical row. Generalizing this approach for such term maps may require a join per row, which is not efficient and is thus best done in the logical table of a triples map.

---

[5]https://github.com/chrdebru/r2rml/tree/r2rml-join

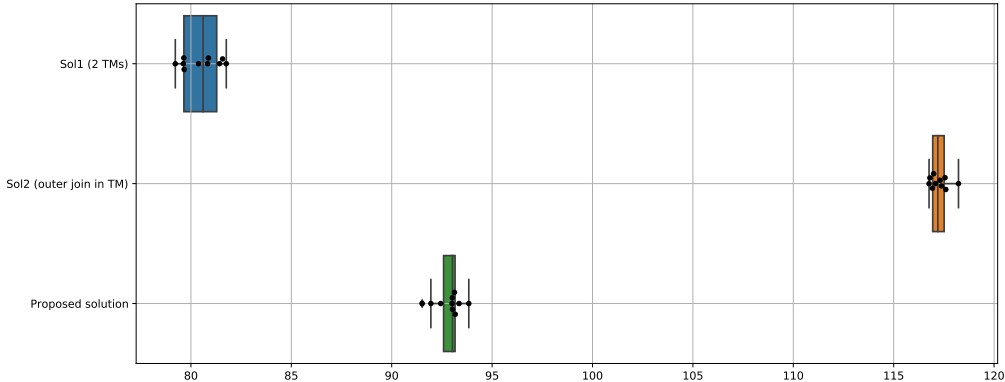

**Figure 2:** Time taken to process three mappings: Sol1–2 triples maps for the outer-join, Sol2– one triples map with the outer-join in the logical table, and out proposed solution.

As we can generate resources with our approach, one can question whether the notion of parent-triples maps is still necessary. The reference algorithm uses both logical tables, even though a processor can only select those used by the subject maps. The question rises: do we refer to (data in) sources, or do we refer to triples maps?

Related to this work is the approach proposed by [8] where they proposed "fields" to manipulate and even combine the source prior to generating RDF. Their work, demonstrated with hierarchical data, aimed to address the problem of references that may yield multiple results and that sources may contain data of mixed formats. They also introduced an abstraction allowing one to retrieve information via a reference that does not depend on the underlying reference formulation. To the best of my knowledge, support for relational databases and the addition of fields from different tables has not yet been published. However, as they declare fields on the logical source, such an approach may boil down to a situation similar to Sol2 mentioned in Section 2.

## 6. Conclusions

We addressed the problem of generating literals from an outer-join, which R2RML does not support. While interesting initiatives are proposed for mostly hierarchical documents, we wanted to address this problem for relational databases by extending R2RML. We proposed a small extension with few implications regarding the R2RML vocabulary. We also extended the reference algorithm and provided an implementation that we have analyzed in an experiment.

From this paper, we can conclude that, for relational databases, our approach is a viable solution. While not as efficient as disjoint triples maps, it may be worth considering not as an approach. It is essential not to consider this vocabulary extension as syntactic sugar, as that would imply it is shorthand for something semantically equivalent. In our approach, the

mappings are self-contained, and the relationship between the two logical tables is thus explicit.

We have addressed this problem for relational databases and R2RML. We could envisage that such an approach could be part of RML, which has the ambition to supersede R2RML. How this approach would work for non-relational data is to be studied.

## A. Mappings Used in the Experiment

```
1   ### MAPPING USED FOR SOL1 IN THE EXPERIMENT
2   <#title_tm>
3       rr:logicalTable [ rr:tableName "title" ] ;
4       rr:subjectMap [ rr:template "http://data.example.com/movie/{id}" ; rr:class ex:Movie; ] ;
5       rr:predicateObjectMap [ rr:predicate ex:title ; rr:objectMap [ rr:column "title" ] ; ] .
6   <#aka_title_tm>
7       rr:logicalTable [ rr:tableName "aka_title" ] ;
8       rr:subjectMap [ rr:template "http://data.example.com/movie/{movie_id}" ; rr:class ex:Movie; ] ;
9       rr:predicateObjectMap [ rr:predicate ex:title ; rr:objectMap [ rr:column "title" ] ; ] .
10
11  ### MAPPING USED FOR SOL2 IN THE EXPERIMENT
12  <#title_tm>
13      rr:logicalTable [
14        rr:sqlQuery "SELECT * FROM title t LEFT OUTER JOIN aka_title2 a ON t.id = a.movie_ID2" ] ;
15      rr:subjectMap [ rr:template "http://data.example.com/movie/{id}" ; rr:class ex:Movie; ] ;
16      rr:predicateObjectMap [ rr:predicate ex:title ; rr:objectMap [ rr:column "title" ] ;
17          rr:objectMap [ rr:column "title2" ] ;
18      ] .
```

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
