# OpenReview forum: "Supporting Relational Database Joins for Generating Literals in R2RML"
_kg-construct.github.io/KGCW/2022/Workshop — KGCW 2022_

### Official Review · ~Herminio_García-González1 · 2022-03-24
**Really well presented paper but it ends up opening even more questions**

**Rating:** 7
**Confidence:** 4

**Review:**

This paper explores the addition of a new construction to the R2RML vocabulary in order to support the generation of literals based on outer-joins. A little experiment on the efficiency of this solution is run and compared with two already existing solutions (workarounds). The main argument for its inclusion in R2RML and RML is that it makes the mapping self-contained and more self-explanatory while not presenting a big drawback on its efficiency.

I find the article well presented and clearly written, which makes it easy to follow. In addition, the idea is worth to be further explored as it can facilitate the generation of literals based on joins for practitioners. However, I find that some points will need a bit more of discussion and it is not totally clear that the main advantage is as advantageous as presented.

It is said that this addition would make the triples map more self-contained and elegant. While this can be widely shared by everybody, I do not feel that Sol1 is as bad as presented. The author assumes that all people using R2RML (or similar tools) will be relational schema minded.  However, I can argue that some people will be thinking more in the output, where, taking into account the RDF compositional property, makes sense to create the two different triples map. In the same way, some people very used to SQL databases (and not with RDF graphs) will by very attracted by Sol2 as it would delegate all the execution and join complexity to the DBMS. Therefore, I would rather not call it a naïve approach but more direct or familiar for SQL practitioners.

About the experiment methodology it is not clear why the mapping is run 11 times. Could be any other number, but the 11 times selection criteria should be further explained. In addition, the mapping could be run a similar amount of times inside each said iteration in order to avoid the mentioned cold run issues and other similar stuff (e.g., garbage collector). Furthermore, it would help to have more stable numbers. As a side note, in Figure 2 the scale unit is missing. Incrementing the iterations to around 30 would also allow to create better comparisons as t-test distribution could be achieved and then stronger comparisons, by means of statistical tests, provided.

The selection criteria for the R2RML engine is not explained. Therefore, one could ask if Sol1 and Sol2 has the same level of efficiency as in the selected R2RML-F engine. Apart from that, the non-normalisation of the used database (including duplicate attributes) draws attention as one could ask if the results for Sol2 would be much different in case of having it normalised.

The author argues that while not as efficient as Sol1, the proposed solution cannot be seen as syntactic sugar as it is not semantically equivalent. We can agree with that, but for the shake of optimisation would it not be worthy to re-write to Sol1 in the engine?

In order to improve the reproducibility of the paper, I encourage the author to share, as supplemental material, the raw measures datasets so other people can confirm the analysis or even take it further.

Other comments and typos per section:

#Introduction

It furthermore proposes [...] This paper furthermore (too many furthermore occurrences used in the same way)

#The Problem

may be biased and real and “messy” → may be biased whereas real and “messy”

but may require the processor to process → it may require the engine to process

#Discussion

To the best of my knowledge → To the best of our knowledge

---

### Official Review · ~Hannes_Voigt1 · 2022-03-27
**Not all possibilities explored**

**Rating:** 7
**Confidence:** 4

**Review:**

The paper proposes an extension to the R2RML mapping language to allow it expressing joins over what ends-up being literals on the RDF end of the mapping. There are two alternative solution discussed in the paper. There alternative solutions not require an extension of R2RML. This is evidence for the fact the extension is not fixing a true fundamental functional deficit in R2RML.

The main argument against Alternative Solution 1 (Sol1) given by the paper seems to be that having two triples maps leads to a bad documentation of the mapping and having to maintain things in two places. It is not obvious why these obstacles couldn't also be overcome by a certain usage patterns, e.g. putting the two related triples maps next to one another in the same code file and add code comments that inform about their relationship.

The main argument against Alternative Solution 2 (Sol2) given by the paper seems to be performance. The performance question, particularly in comparison to the proposed third solution, seems to come down to where the join is done — in the mapper or in the RDBMS — and how much redundancy there is in the data that gets shipped from the RDBMS to the mapper. It is hard to believe that the mapper has a fundamental advantage in computing the join, at least the paper does not provide any indication in that direction. Hence, the performance difference seems to primarily result from the redundancy in the joined data.

This suggests a forth possibility: Using nested data/NF^2 capabilities at the RDBMS end to have a factored representation of the joined data. In the paper's example, each title record have a list of its aka_titles. The SQL standard and various RDBMS (maybe not MySQL) have the capability to compute such results, either with SQL arrays or with JSON. Now, unpacking a list of values into multiple triples might also require an extension to R2RML. R2RML extension for dealing with JSON data buried in relational table do already exist, however. Hence, it is unforunate that the paper does not explore this forth possibility and at least compare the proposed solution to it.

Well, all of that gives good material for a discussion at the workshop.

Editorial comments

 * Sec 1, "R2RML was conceived for relational databases, but can be applied to relational data." This sentence is somewhat confusing. In which respects is relational data signifanctly different from the data in relational databases, such that it is a surpise that something developed for relation databases also works for relational data?
 * Sec 2, "In short, their motivation was that existing synthetic benchmarks [that] may be biased and real and “messy” provided better grounds for comparison." seems to miss the word "that".
 * Footnote 2, "... values in aka_table were represented ...": The running text only mentions a table named aka_title. What is aka_table?
 * Footnote 2, "The tables contain 2528312 and 361472 respectively.": Do the tables literally contain these two numbers? Or do they contain that many instance of something, which is not mentioned? It is records, fields, values, bytes, omlettes or pancakes?
 * Sec 2, "There are two approaches to solving this problem with R2RML:" This sentences point to a problem which just has been introduced – that is how the sentence reads at least. Unfortunately, no problem has been introduced in the previous paragraph. Particular, no problem regarding the movie data introduce in the previous paragraph.

---

### Official Review · ~Jhon_Toledo_Barreto1 · 2022-03-30
**Good paper, and it adds valuable discussions about joins on R2RML.**

**Rating:** 7
**Confidence:** 4

**Review:**

This paper proposes a simple extension of the R2RML vocabulary and extends the reference algorithm for generating RDF triples to support the creation of literal based on outer joins. The author used the Internet Movie Database benchmark to measure the performance of the joins. The paper is well written and easy to read overall.

In section 1&2, the author positions his work by proposing three solutions, Sol1, Sol2 and the proposed solution(Sol3), and describes clearly their limitations and advantages.
He starts mentioning that he uses "fairly big" relational databases for IMDb benchmarking joins. The statement is a bit subjective IMO. The "fairly big" depends not only on the number of rows, but on the parameters that can affect the construction of a KG (see [1]). Fortunately, the author describes the characteristics of his experiments such as the tables, records, foreign keys, joins. Those characteristics determine whether relational databases are "big" or "small" for this specific use case.

In section 4, I've some comments and questions:

* The author describes a little experiment comparing Sol1, Sol2 and Sol3. The number of ran execution times (11) is arbitrary, is this motivated on any benchmark?.Whether the experiment is extended, do you think is necessary to add another benchmark?.
* Using the mapping of Sol1, some engines retrieve data from a database selecting only the columns referred to in the triples maps. In Sol2, into the mapping, the "logicalTable" shows a SQL query. In Sol3 does it select only the columns referred to in the mapping?, is there any research or optimisation about that in Sol3?.
* The Sol3 retrieves data of all titles using outer-join with variations in the title (aka_title), this solution could answer which titles have variation titles(left-join)?. In his approach exists the possibility of considering questions like, which variations of title do not have a "main" title(right-join)? or which titles exactly have variation(maybe with full join)?.

In the evaluation would have been interesting to see a more gradual evaluation (records and outer-joins) and to see the trend of the results and maybe remove the foreign key. For sure the paper will spark some discussions during the workshop and I would like to see more implementations and approaches that solve this problem (as well as benchmarks, testbeds, etc.)

The contributions of the paper are the following:

* Proposing self-container mappings in R2RML to create explicit relations between TriplesMap.
* This paper extends the generated RDF triples algorithm.
* In the R2RML context, the author adds an important discussion on whether we should refer to data sources or TiplesMaps.
* Add valuable discussions about joins in R2RML mappings.

Other comments:

* Although section 4 mentions that figure 2 shows the average time in seconds, is recommended to put also the time in the axes of the figure.
* Does the author consider the possibility of integrating Inner-join?.



[1] Chaves-Fraga, D., Endris, K.M., Iglesias, E., Corcho, O., Vidal, ME. (2019). What Are the Parameters that Affect the Construction of a Knowledge Graph?. In: Panetto, H., Debruyne, C., Hepp, M., Lewis, D., Ardagna, C., Meersman, R. (eds) On the Move to Meaningful Internet Systems: OTM 2019 Conferences. OTM 2019. Lecture Notes in Computer Science(), vol 11877. Springer, Cham. https://doi.org/10.1007/978-3-030-33246-4_43

---

### Official Review · ~Pano_Maria1 · 2022-04-01
**Interesting proposal, that begs more exploration**

**Rating:** 7
**Confidence:** 4

**Review:**

This paper introduces an extension to referencing object maps in R2RML to afford the generation of literals from outer joins.
The paper is clearly written and well presented.

The author compares existing approaches for generating literals from an outer join using R2RML and introduces a new approach. The criteria used are self containedness of the mapping rules and performance of several implementation approaches.

The comparison with the existing approaches nicely shows the performance impact of the solutions, however, the proposed solution is presented as if designed based only on the use case of generating literals from outer joins where the join is made to generate triples for the same resource type.

An example of generating literals from an outer join, whose parent table is also used to generate triples for another resource type, is not discussed. This raises the question what impact that use case would have on the proposed language extension and the proposed extension to the reference algorithm, again taking the same criteria into account.

In any case this proposal is an interesting discussion piece for this workshop.

---

### Decision · Program_Chairs · 2022-04-11

**Decision:**

Accept

**Comment:**

Dear author,

Thank your for submitting your paper. We are happy to inform you that we accept your paper! Please carefully consider the reviews when you prepare your paper for the camera-ready version. You will receive specific instructions to submit your camera-ready soon.

Kind regards
Organizers of the Knowledge Graph Construction workshop 2022